# Medical Scientific Table-to-Text Generation with Synthetic Data under Data Sparsity Constraint

**Heng-Yi Wu**[*]
Development Science Informatics
Genentech Inc
wu.hengyi@gene.com

**Jingqing Zhang** [*]
Pangaea Data Limited
jzhang@pangaeadata.ai

**Julia Ive** [†]
Pangaea Data Limited
jive@pangaeadata.ai

**Tong Li**
Pangaea Data Limited
tli@pangaeadata.ai

**Vibhor Gupta**
Pangaea Data Limited
vgupta@pangaeadata.ai

**Bingyuan Chen** [‡]
Development Science Informatics
Genentech Inc
chenb14@gene.com

**Yike Guo** [‡ §]
Pangaea Data Limited
yguo@pangaeadata.ai

## Abstract

An efficient table-to-text summarization system can drastically reduce manual efforts to understand and summarise tabular data into textual reports. However, in practice, the problem is heavily impeded by data sparsity and the inability of the state-of-the-art natural language generation models (such as T5, PEGASUS, and GPT-Neo) to produce coherent and accurate outputs. This is particularly true in pre-clinical and clinical domains. In this paper, we propose a novel table-to-text approach and tackle these problems with the help of synthetic data generation as well as copy mechanism. Experiments show that the proposed method can boost the performance of copying concise and relevant information from tabular data to generate assay validation and toxicology reports.

## 1 Introduction

Neural table-to-text (table2text) generation, which aims to condense tabular data into textual narratives automatically, has been studied with various successful approaches (2; 4; 19). In the medical domain, an automated table2text system can support clinical scientists in accelerating the writing of medical reports based on experimental tabular data with time and cost savings.

However, neural table2text generation of the medical and scientific text still remains a challenging problem (10). Data sparsity is the main barrier, as scientific medical summaries are produced in small quantities with high-level quality by domain experts however may lack adequate numbers of extractable observations. In addition, the factual consistency between the tabular values and the text is essential for the validity of the generated text. The state-of-the-art text generation models, especially

---

[*]Heng-Yi and Jingqing contributed equally to this work.

[†]This work was completed when Julia was at Pangaea Data Limited and Julia now works at Queen Mary University of London.

[‡]Bingyuan and Yike are corresponding authors.

[§]Yike also works at Imperial College London and Hong Kong Baptist University.

NeurIPS 2022 Workshop on Synthetic Data for Empowering ML Research.

Transformer based (17) models, lack the ability to learn how to copy the relevant values accurately from the input tables to the text narratives in a low resource setting, which limits these models from generating high-quality reports.

To address the aforementioned challenges, we propose a two-step architecture with synthetic data generation, and a copy mechanism (14; 18) to increase the accuracy of the table2text model in presenting key tabular values in generated textual narratives.

The proposed architecture consists of two modules, the table extractor to extract the most relevant and principal values from input tables and the text generator to generate textual narratives based on the extracted values. Both modules are guided by copy mechanism, which acts as pointers to directly copy principal tabular values to the generated text. To ensure that relevant and principal values remain consistent between input tables and generated text, we propose a synthetic data generation method that creates training samples by slot-value replacements respecting the type of values (1). For example, "200 ml" will be replaced by similar values with similar conditions, such as "100 ml". The proposed synthetic data generation promotes training examples where only principal tabular values are changed. The copy mechanism allows the table extractor and text generator to focus on these principal tabular values and ensure the accuracy of copying.

We collect two datasets to train and evaluate our methods, namely assay validation reports and toxicology reports. Experiments show that our approach with synthetic data generation achieves state-of-the-art performance compared to baseline methods. In summary, the main contributions of our work are: (1) we propose a table2text architecture for the generation of accurate medical scientific reports; (2) we propose a method of synthetic data generation with copy mechanism to address data sparsity for the medical scientific table2text.

## 2   Datasets

In this study, we collect two datasets from anonymous provider, namely assay validation reports and toxicology reports, to train and evaluate our proposed methods. Table 1 shows the statistics of the two datasets, and examples from the two datasets are provided in Appendix.

**Assay Validation Reports**   are written to describe the quantitative performance of an assay, including its accuracy, sensitivity, specificity, precision, detection limit, range, and quantitation limits. In addition, complete reports contain inter-assay and inter-laboratory assessments of assay repeatability and robustness. In this study, we collect pairs of tables and paragraphs from 92 raw complete assay validation reports. These pairs are constructed automatically by firstly matching the table number and then selecting the most relevant paragraph that contains most of the values from the table. Overall, 1,239 pairs are collected, which are split into the training set (1,133 pairs) and testing set (106 pairs).

**Toxicology Reports**   are expert statements describing the results of pre-clinical toxicology studies carried out by pharmaceutical companies. These documents have been identified as valuable sources of safety findings for investigational drugs. In this study, we focus on the summarized findings of body weight changes, clinical observations, and mortality rates. We manually collect 87 tables and paragraph pairs, split into the training set (43 pairs) and testing set (44 pairs).

Table 1: Statistics of assay validation reports and toxicology reports.

|  | Assay Validation Reports (Training) | Assay Validation Reports (Testing) | Toxicology Reports (Training) | Toxicology Reports (Testing) |
|---|---|---|---|---|
| #, pairs of tables and reports | 1133 | 106 | 43 | 44 |
| avg #, tokens in tables | 307.0 | 308.6 | 62.5 | 73.7 |
| avg #, tokens in reports | 126.8 | 170.0 | 43.8 | 53.0 |

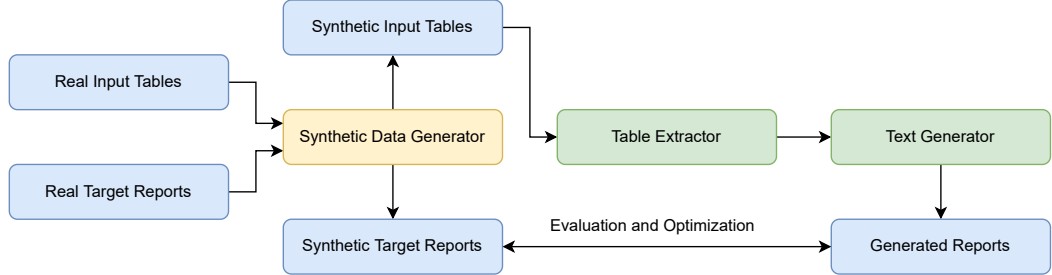

Figure 1: The architecture with table extractor, text generator, equipped with Synthetic Data Generator.

# 3 Methodology

## 3.1 Architecture

We formalize the table2text problem as a sequence-to-sequence problem; hence the input tables are flattened row by row into a sequence of tokens, whereas the target reports are already in free text. As shown in Figure 1, following best practices in the domain (12), we propose a two-step architecture with a table extractor and a text generator to generate scientific medical narratives from tabular data. Figure 2 shows that **table extractor** takes the full input table and extracts the most relevant values. The extracted values contain key values from the tables in the order that they should appear in the target text. The key values are usually biomedical concepts (such as drug names or disease names extracted from the real training samples), identifiers of experiments, abbreviations, and numerical values. The extracted values only focus on the valuable information concerning the target during training and ignore other characters such as punctuation marks or common words. The extractor is trained to capture repetitions if the values are repeated in the target text and maintain the order of these values with respect to other key values. Figure 3 shows that **text generator** takes the extracted values from table extractor and generates the respective medical scientific narratives. We also prepend the titles of all input tables and the last three rows of each full table, which provide additional information and strengthen the generation capability of the text generator.

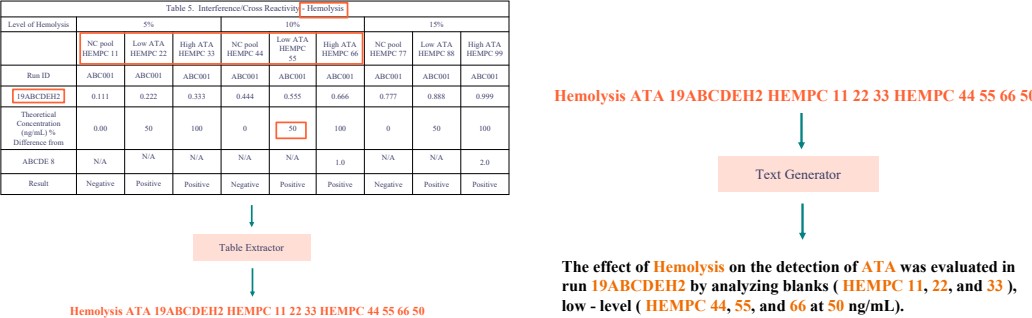

Figure 2: The table extractor takes the full table as the input and extracts key tabular values.

Figure 3: The values extracted from the table extractor are fed into the text generator to generate the output text.

As inputs and outputs are both sequences of tokens, both table extractor, and text generator are implemented by Transformer encoder-decoder architecture, which is guided by copy mechanism following (6). The copying mechanism incorporates a switching function to decide whether to generate a word by the decoder or to copy an crucial word from the input directly. The decision is monitored by the attention distribution over the encoder representations at each decoder timestamp.

## 3.2 Synthetic Data Generator

As the table2text generation task is defined as a two-step process of extracting principal values from tables and describing these values with textual narratives, the values extracted from the table and

copied into the text should be matched to preserve factual consistency. Therefore, we follow the previous study (1) to propose a slot-value replacement approach. Each slot is filled with a random value as long as the type of value (such as integers, floats, strings) at this slot stays the same. For example, a string can only be replaced by another string. To create pairs of synthetic tables and reports, if a value in the input table is replaced by a new value, the corresponding value in the target report will also be replaced.

**Assay Validation Reports**   The following value types are considered for assay validation reports, namely drug names, disease names, biomedical abbreviations (such as ADA for anti-drug antibody, CV representing coefficient of variation), integers, floats, identifiers of experiment runs, and identifiers of tables. The drug names, disease names, and biomedical abbreviations are replaced by other values of the same type from a dictionary. For example, "Hemolysis" can be replaced by "Lipolysis". Numerical values are created by randomizing the original values but ensuring that the new values are close to the original values (such as using 0.71 to replace 0.70). Alphanumerical values (such as the identifiers of experiments and tables) can be created by randomizing the order of the original values (such as "CDE123" modified to "C1D2E3"). Figure 4 in Appendix provides an example of the synthetic data.

**Toxicology Reports**   Slightly different from assay validation reports, the following value types are defined for toxicology reports: drug names, genders, integers, and floats. Drug names (such as "G1234") can be replaced by either randomly picking up a value from the collected dictionary for drugs (such as "G1235") or randomizing the order of the original values (such as "G4321"). Genders are represented by either "F" or "M", and the genders in the real data are swapped to the opposite genders. Numerical values, such as the dosage level, can be created by randomizing the original values slightly. For the findings of clinical observations, we build a dictionary that contains all existing clinical observations, such as hair loss, swelling, and crust. The original observation can be replaced by randomly selecting another observation from the dictionary. For the findings of mortality rates, one additional dictionary is built to indicate the status of the animals (such as terminal sacrifice, found dead, moribund sacrifice). The status of the synthetic samples can be randomly selected from this dictionary. An example of the synthetic data is provided by Figure 5 in Appendix.

For each pair of real tables and reports, we generate 1,000 synthetic pairs. Combined with the copy mechanism, our synthetic samples generated by token-level modification promote precision of copying relevant values from inputs to the outputs, as these token-level differences between training samples force the model to learn the importance of these values.

## 4   Experiments

### 4.1   Implementation Details

The code was implemented by Python 3.8, and Pytorch (9). Both models (table extractor and text generator) weights were initialised from `PEGASUS-pubmed` (21) checkpoint from transformers library (20). We fine-tuned each model for 50,000 steps with a batch size of 4, and Adafactor (15) as the optimizer.

### 4.2   Baseline Methods

We compared our approach with other Transformer based models, which directly took tables as inputs to generate reports in a single step rather than two steps. (1) **PEGASUS** (21) is a Transformer encoder-decoder model which achieves state-of-the-art performance on 12 abstractive summarisation datasets. We fine-tuned `PEGASUS-pubmed` checkpoint from the transformers library for 30,000 steps with a batch size of 4. (2) **T5** (13) is also a Transformer encoder-decoder which unifies natural language processing tasks into a text-to-text format and defines a multi-task mixture of unsupervised and supervised tasks as pre-training objectives. We fine-tuned `t5-large` checkpoint from the transformers library for 30,000 steps with a batch size of 4. To achieve optimal results for T5, we updated the length penalty to 1.2 to encourage the model to generate longer sequences. (3) **GPT-Neo** is an open-sourced GPT model which was pre-trained on the Pile corpus (3) and fine-tuned for 30,000 steps using the teacher-forcing method to use the ground truth from a prior time step as input. In addition to Transformer based models, we also considered (4) **content selection and**

**planning** (11) which separated the generation process into two stages. The first stage produced a content plan highlighting the order of the values to appear in the text and the second stage took the content plan to generate the report using a Long Short-Term Memory (LSTM) (5) encoder-decoder model. We constructed the gold-standard content plan by extracting the overlap tokens between input tables and target reports. (5) **Template-based algorithm** was designed to find the closest report from the real training set as a template for report generation. For each table, we selected the closest template according to table titles using the longest contiguous matching sub-sequence algorithm. The token-level slots in the selected template were filled with values from the corresponding positions in the test table.

## 4.3 Evaluation Metrics

Considering the importance of copying the tabular values from the input tables to the generated reports with high precision, we have used two metrics focused on the assessment of copying these values.

Given the unique tabular values from input tables that appear in target reports, **Table Recall** estimates the percentage of such unique tabular values that actually appear in the generated reports. The order and count of the tabular values are not taken into consideration.

**BLEU Extract** computes the precision values of consecutive token spans between the table extract restored from the generated report and the reference table extract. The table extracts restored from the text contain only the words and numbers present both in the generated text and in the reference table extract. This metric is proposed based on the popular text generation metric BLEU (8). In this metric, the order and count of the tabular values from the tables are considered.

Apart from the above metrics, standard metrics including ROUGE (7), BLEU (8) and TER (16) for comparing lexical content of target and generated text are also reported. ROUGE measures the n-grams overlap between target text and generated text, BLEU focuses on the n-gram precision, and TER takes the minimum number of edits into consideration and quantifies the efforts required to change a generated text to its target, so the lower TER means the generated text is better.

## 4.4 Quantitative Results

Table 2: Results on Assay Validation Reports. Please note all methods in this table do not use synthetic training data unless it is specified.

| | Table Recall ↑ | BLEU Extract ↑ | ROUGE 1 ↑ | ROUGE 2 ↑ | ROUGE L ↑ | BLEU ↑ | TER ↓ |
|---|---|---|---|---|---|---|---|
| Template | 0.3856 | 27.07 | 0.5666 | 0.4192 | 0.4984 | 40.30 | 0.6661 |
| (11) | 0.5352 | 20.38 | 0.3996 | 0.2318 | 0.3140 | 17.34 | 1.4980 |
| GPT-Neo | **0.8577** | 23.82 | 0.2110 | 0.0715 | 0.1207 | 1.83 | 1.3508 |
| T5 | 0.4923 | 15.87 | 0.3968 | 0.1956 | 0.2435 | 14.82 | 1.1228 |
| PEGASUS | 0.5927 | 39.49 | 0.6128 | 0.4927 | 0.5550 | 42.04 | 0.7147 |
| Ours w/o Synthetic Data | 0.6497 | **47.61** | **0.6591** | **0.5422** | **0.6029** | **48.84** | **0.6470** |
| Ours with Synthetic Data | **0.7245** | 46.92 | 0.6590 | 0.5304 | 0.5986 | 44.36 | 0.6455 |

Table 3: Results on Toxicology Reports. Please note all methods in this table are enhanced with synthetic training data.

| | Table Recall ↑ | BLEU Extract ↑ | ROUGE 1 ↑ | ROUGE 2 ↑ | ROUGE L ↑ | BLEU ↑ | TER ↓ |
|---|---|---|---|---|---|---|---|
| GPT-Neo | 0.9127 | 25.03 | 0.6980 | 0.2867 | 0.5132 | 8.72 | 1.2586 |
| T5 | 0.8784 | 26.47 | 0.7364 | 0.5874 | 0.7240 | 41.79 | 0.5990 |
| PEGASUS | **0.8999** | **29.98** | 0.9038 | **0.7799** | 0.8867 | 64.71 | 0.2789 |
| Ours | 0.8897 | 29.49 | **0.9055** | 0.7760 | **0.9034** | **65.18** | **0.2640** |

For assay validation reports, Table 2 shows that our method, without using synthetic data, outperforms other baseline methods on all evaluation metrics. By comparing our method with other Transformer models, it suggests that the two-step architecture with table extractor and text generator is more

effective than a single-step Transformer model. Also, combined with synthetic training data, the proposed method achieves significantly higher Table Recall (0.7245) than the baseline methods. This indicates that using synthetic data can be useful to improve the accuracy of copying key values from inputs to outputs.

For toxicology reports, due to the limited number of real training samples, the baseline methods can not produce reasonable results for evaluation without synthetic data; hence all baseline methods, as well as the proposed approach, are trained with synthetic data. As shown in Table 3, with synthetic data, PEGASUS and our method achieves comparable scores, which are higher than T5. As both PEGASUS and our method are initialised by the PEGASUS-pubmed checkpoint, it may indicate that pre-training a model on PubMed is beneficial for scientific medical applications and a single-step model can be sufficient to generate good results for toxicology reports.

However, interestingly, GPT-Neo tends to simply copy the input tables into the output, which achieves an exaggerated Table Recall score of 0.8577 for assay validation reports and 0.9127 for toxicology reports, but other scores are rather low.

## 5    Conclusion

In this paper, we propose a table2text method with a two-step architecture, including a table extractor and text generator, to generate scientific medical reports from tabular data. Coupled with copy mechanism, we propose synthetic data generation, which is shown to enhance the performance of the model. Experiments show the proposed method achieves state-of-the-art performance in generate assay validation reports and toxicology reports compared with baseline methods. With the precise value extraction from the tables and the concise text generation for drafting a report, it can dramatically reduce manual efforts in writing prospective reports in the scientific medical domain and speed up regulatory report submission for pre-clinical and clinical studies.

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

# 6 Appendix

| Table 5. Interference/Cross Reactivity - Hemolysis | | | | | | | | | |
|---|---|---|---|---|---|---|---|---|---|
| Level of Hemolysis | 5% | | | 10% | | | 15% | | |
| | NC pool HEMPC 11 | Low ATA HEMPC 22 | High ATA HEMPC 33 | NC pool HEMPC 44 | Low ATA HEMPC 55 | High ATA HEMPC 66 | NC pool HEMPC 77 | Low ATA HEMPC 88 | High ATA HEMPC 99 |
| Run ID | ABC001 | ABC001 | ABC001 | ABC001 | ABC001 | ABC001 | ABC001 | ABC001 | ABC001 |
| 19ABCDEH2 | 0.111 | 0.222 | 0.333 | 0.444 | 0.555 | 0.666 | 0.777 | 0.888 | 0.999 |
| Theoretical Concentration (ng/mL) % Difference from | 0.00 | 50 | 100 | 0 | 50 | 100 | 0 | 50 | 100 |
| ABCDE 8 | N/A | N/A | N/A | N/A | N/A | 1.0 | N/A | N/A | 2.0 |
| Result | Negative | Positive | Positive | Negative | Positive | Positive | Negative | Positive | Positive |

| Table 5. Interference/Cross Reactivity - Lipolysis | | | | | | | | | |
|---|---|---|---|---|---|---|---|---|---|
| Level of Lipolysis | 5% | | | 10% | | | 15% | | |
| | NC pool HEMPC 12 | Low ATA HEMPC 23 | High ATA HEMPC 34 | NC pool HEMPC 45 | Low ATA HEMPC 56 | High ATA HEMPC 67 | NC pool HEMPC 77 | Low ATA HEMPC 88 | High ATA HEMPC 99 |
| Run ID | ABC001 | ABC001 | ABC001 | ABC001 | ABC001 | ABC001 | ABC001 | ABC001 | ABC001 |
| ABCDEH192 | 0.111 | 0.222 | 0.333 | 0.444 | 0.555 | 0.666 | 0.777 | 0.888 | 0.999 |
| Theoretical Concentration (ng/mL) % Difference from | 0.00 | 50 | 100 | 0 | 51 | 100 | 0 | 50 | 100 |
| ABCDE 8 | N/A | N/A | N/A | N/A | N/A | 1.0 | N/A | N/A | 2.0 |
| Result | Negative | Positive | Positive | Negative | Positive | Positive | Negative | Positive | Positive |

Real Target:

The effect of Hemolysis on the detection of ATA was evaluated in run 19ABCDEH2 by analyzing blanks ( HEMPC 11, 22, and 33), low - level ( HEMPC 44, 55, and 66 at 50 ng/mL).

Synthetic Target:

The effect of Lipolysis on the detection of ATA was evaluated in run ABCDEH192 by analyzing blanks ( HEMPC 12, 23, and 34), low - level ( HEMPC 45, 56, and 67 at 51 ng/mL).

Figure 4: Example of synthetic values created from the original table for assay validation reports. In this example, "Hemolysis" is replaced by "Lipolysis". The synthetic value "ABCDEH192" is created by randomising the original value "19ABCDEH2". Numerical values such as "50" have been modified to new values "51" that are close to the original values.

| Sex | Group Set | Subject ID | 1-7 | 1-8 |
|---|---|---|---|---|
| F | 1--Vehicle | Mean | -2.00 | -0.01 |
| | 2--100mg/kg ABCD0001 | Mean | -1.01 | -0.02 |
| | 3--200mg/kg ABCD0001 | Mean | -3.00 | -0.03 |
| | 4--300mg/kg ABCD0001 | Mean | -2.02 | -0.04 |
| M | 1--Vehicle | Mean | 10.01 | 0.05 |
| | 2--100mg/kg ABCD0001 | Mean | 10.03 | 0.06 |
| | 3--200mgkg ABCD0001 | Mean | 10.04 | 0.07 |
| | 4--300mg/kg ABCD0001 | Mean | 0.01 | -0.08 |

| Sex | Group Set | Subject ID | 1-7 | 1-8 |
|---|---|---|---|---|
| M | 1--Vehicle | Mean | -2.02 | -0.01 |
| | 2--200mg/kg ABC0001D | Mean | -1.01 | -0.02 |
| | 3--300mg/kg ABC0001D | Mean | -3.15 | -0.03 |
| | 4--400mg/kg ABC0001D | Mean | -2.02 | -0.04 |
| F | 1--Vehicle | Mean | 10.05 | 0.05 |
| | 2--300mg/kg ABC0001D | Mean | 10.03 | 0.06 |
| | 3--400mgkg ABC0001D | Mean | 10.04 | 0.07 |
| | 4--500mg/kg ABC0001D | Mean | 0.02 | -0.08 |

Real Target:

Body weight change consisting of **1.00**% loss compared to control in **females** at **200.0 mg/kg**; Body weight change consisting of **10.00**% loss compared to control in **males** at **300.0 mg/kg**

Synthetic Target:

Body weight change consisting of **1.13**% loss compared to control in **males** at **300.0 mg/kg**; Body weight change consisting of **10.03**% loss compared to control in **females** at **500.0 mg/kg**

Figure 5: Example of synthetic values created from the original table for toxicology reports (the findings of body weight changes). In this example, the original drug name "ABCD0001" has been modified to "ABC0001D". Dosage levels such as "200mg/kg" are replaced by other values such as "300mg/kg". The percentages of body weight changes are also replaced by randomising the original values slightly.

