# OpenReview forum: "Medical Scientific Table-to-Text Generation with Synthetic Data under Data Sparsity Constraint"
_NeurIPS.cc/2022/Workshop/SyntheticData4ML — Neurips 2022 SyntheticData4ML_

### Official Review · Reviewer_SFDG · 2022-10-16
**Simple and effective method**

**Rating:** 8
**Confidence:** 4

**Review:**

This paper presents a two-stage approach for table-to-text generation task: first extract the related cell values in the table, then convert these cell values into the final text narratives, both stages are realized with a Transformer encoder-decoder model. During the two-stage process, slot-value replacement is applied for synthetic data generation, the experiments on two medical datasets demonstrate the effectiveness of the propose method.
The overall method is very simple but also effective, especially in the low-resource setting of medical domain. The comparison experiments and evaluations are comprehensive, it's interesting to see the synthetic data can improve the table recall significantly but not lower the generated text quality, overall, this is a good work.

---

### Official Review · Reviewer_jJsM · 2022-10-17
**Overall, I think the paper made some contributions but the novelty of the work needs contexts of related work and additional explanations and experiments are needed to back up the claims made in the paper**

**Rating:** 6
**Confidence:** 4

**Review:**

This paper considers the table-to-text task in the medical domain. The authors utilized a two-step procedure (first extracting important values and second transforming values to text). They applied copy mechanisms to ensure the alignment between values in the table and the generated text and created synthetic data by swapping the values in the table and texts simultaneously.
They found the proposed methods outperformed/were comparable to the other baselines in several metrics across two datasets and synthetic data could help with the table recall while maintaining relatively good language quality in one dataset.

Summary of Strength:
1. Their work addresses an important problem that has useful downstream applications (table2text, medical domain, low-resourced)
2. The general presentation of the paper is clear
3. They collected medical datasets for testing

Summary of Weakness:
1. The current paper does not review recent related table-to-text works and thus the novelty of the work is discussed out of context.
    There are quite a number of recent table-to-text works that utilize a two-stage strategy, data augmentation methods. Here are some examples:
     https://arxiv.org/pdf/1908.03067.pdf
     https://arxiv.org/pdf/2102.03556.pdf
    https://arxiv.org/pdf/2005.00969.pdf
2. while it is insightful that authors tested their methods in new medical datasets, I wonder whether the method can be applied to and evaluated on public datasets such as wiki-bio. The performance on a public dataset can help researchers who work in the field to better assess and replicate the proposed method.
3. Along the line of evaluation and performance, according to the paper I think the two main metrics are table recall and BLEU Extract. In the first dataset, GPT-Neo has the best Table-Recall performance and in the second dataset, Pegasus has the best scores in both table-recall and bleu-extract metrics compared to the proposed method. I wonder to what extent these results affect the claims that the proposed two-stage architecture is more effective and also the proposed method outperforms the other baselines.
4. In addition, the current method differs from other baselines in two main ways (two-step + copy mechanism), it would be difficult to attribute the performance gain to a specific aspect. Maybe additional ablation studies such as using the current model but conducting a one-step procedure can help the audience to better understand the source of improvement.

While the general structure is clear, there are a few places that might need some clarifications or modifications:
1. Paragraph (27-36) is a bit confusing because lines (27-31) introduce the model structure and lines (31-35) talk about synthetic data, and then lines (35-36) talk about copy mechanisms. It is a bit confusing how the synthetic data is related to the copy mechanisms. My understanding is that the table-to-value extractor and the value-to-text generator utilize the copy mechanism but the synthetic data is an independent component.
2. In the experiment sections: the description of the current model architecture is not clear to me. Is it a transformer-based model with a copy mechanism implemented in the decoder? The implementation details mentioned the PEGASUS-PubMed so is it essentially pegasus + copy_mechanism? Adding more details to the descriptions of the model architecture of the current model can help to clarify the confusion.

---

### Official Review · Reviewer_AVFG · 2022-10-18
**Relevant subject, interesting approach, details missing, some overlooked areas, some concerns about the semantic quality of the generated results**

**Rating:** 6
**Confidence:** 3

**Review:**

This paper proposes a method to generate textual reports from tabular data. Although not explicitly mentioned, the paper seems to be focused on medical domain applications. The method aims to solve the problem of lack of sufficient training instances. They define the main challenge for the model to be copying important values from the table to the report accurately. The proposed method has an extractor component a table generator component. The problem is defined as a sequence-to-sequence task. The extractor extracts important values from the table and transforms them into a sequence. The synthetic data generator does slot-value replacement to generate additional data. The report generator uses this synthetic dataset to generate relevant reports.

The work is well motivated and certainly beneficial in certain domains such as medical. The paper presents an interesting approach. It could be further improved by providing more details about the mechanisms and making clarifications and justifications in terms of the semantic aspects of the generated report, addition of more metrics, extending the experiments and providing more detailed results.


Remarks:
- The paper lacks details on the actual architecture/implementation/hyperparameters of the model. Also the terminology is a bit confusing. There is emphasis on the importance of the "copy" mechanism but nowhere in the paper that mechanism is clearly defined. One could guess what the "copy" unit is in the presented pipeline, but still this process should be clearly defined.
- My understanding is that tabular medical results could get quite complicated, with merged columns and references to other cells. I also gather some values could be interpreted in connection with other values, in which case they probably should either be mentioned in the same paragraph/sentence/etc. or be referenced explicitly. I don't see how the model handles this kind of complicated structure and semantic dependency.
- Based on the example provided in fig. 3 it is clear that the extractor is guided during the training by the generated report in assessing which values in the input table are "important". This could be a point of concern. For example, if another expert would have written the same report for the same table would they also use the same order in reporting the results? Would they include more or less values from the table? What if we have reports from different experts on the same type of table with different styles of writing. What if they refer to something that is common knowledge by clinicians but is outside the included information in the table?
- How does the model generalize? What if the order of columns of the input table are changed? Would the model still be able to extract the same principal values from the table with the same order as to not confuse the report?
- The slot value replacement is also unclear. Would the generated values still fall within the acceptable bounds? I don't see any semantic correctness check. If not, wouldn't it compromise the report generator? I assume a systolic blood pressure of 120 and a much higher value would have different interpretations. So if the values are changed completely at random then there might be a chance that important signals in the table are being randomly turned on and off.
- An important step in preprocessing of the data is mentioned as replacing abbreviations with others using a dictionary. Is this step necessary for all dataset/reports? What would happen if the dictionary does not exist and the terminology changes between systems/hospitals/experts. If the performance of the model suffers when this step cannot be performed, it should be clearly mentioned as a prerequisite.
- The defined metrics are helpful. However the table recall metrics seems to be missing the semantic aspects of the report, e.g. clearly forming sentences linking concepts to values, capturing indirect references, etc.
- Why results for toxicology reports are reported only with synthetic data, but for assay validation both w and w/o synthetic data are reported?
- Looking at the results for assay validation - bold numbers in the table-, it seems like adding the synthetic data component does not make a significant contribution to the model's performance. How do the authors justify the necessity for it then?

---

### Meta-Review · Area_Chair_fTRv · 2022-10-18

**Recommendation:** Accept